# Perinatal and Neonatal Outcomes in Fetal Growth Restriction and Small for Gestational Age

**DOI:** 10.3390/jcm11102729

**Published:** 2022-05-12

**Authors:** Chiara Lubrano, Emanuela Taricco, Chiara Coco, Fiorenza Di Domenico, Chiara Mandò, Irene Cetin

**Affiliations:** 1Department of Woman Mother and Neonate ‘V. Buzzi’ Children Hospital, ASST Fatebenefratelli Sacco, 20154 Milan, Italy; emanuela.taricco@asst-fbf-sacco.it (E.T.); chiara.coco@unimi.it (C.C.); fiorenza.didomenico@unimi.it (F.D.D.); irene.cetin@unimi.it (I.C.); 2Department of Biomedical and Clinical Sciences, University of Milan, 20157 Milan, Italy; chiara.mando@unimi.it

**Keywords:** fetal growth restriction, SGA, blood gas analysis, perinatal outcomes

## Abstract

Alterations in intrauterine fetal growth increase the risk of adverse perinatal and neonatal outcomes. In this retrospective study, we analyzed data of 906 pregnancies collected in our maternal fetal medicine center, with different patterns of growth: 655 AGA (Appropriate for Gestational Age), 62 SGA (Small for Gestational Age: fetuses born with a weight less than 10° centile, not diagnosed before delivery), 189 FGR (Fetal Growth Restriction, classified in early and late according to gestational week at diagnosis). For each group, we compared maternal characteristics, gestational age at delivery, and perinatal and neonatal outcomes. Risk factors for fetal growth alterations were advanced age, being primiparous, and a lower pregestational BMI. FGR fetuses were born at earlier gestational ages (32 [IQR 29–38] early-FGR and 38 [IQR 36–39] late-FGR), with blood gas values comparable to the AGA group but worse neonatal outcomes related to prematurity. Unexpected SGA fetuses born by vaginal delivery, managed as AGA, were more hyperlactacidemic (4.4 [IQR 2.7–5.5]) and hypoxemic (−5.0 [IQR −7.1–2.8]) at birth than both AGA and FGR. However, neonatal outcomes (accesses and days of hospitalization in NICU) were better than FGR, likely due to gestational age and birthweight similar to AGA.

## 1. Introduction

Fetal growth restriction (FGR) occurs when the fetus does not reach its biological growth potential. It affects approximately 7–15% of pregnancies [1,2,3], with a mortality of 12% in the fetal period and 8% in the neonatal period [4].

Approximately 60% of FGR cases are idiopathic and multifactorial, while, in 40% of cases, a definite etiology is recognized (pre-existing maternal diseases, chromosomopathies, fetal malformations, infectious diseases) [5].

FGR secondary to placental insufficiency is associated with an incomplete vascular adaptation of the uterine circulation. This leads to a reduction of uteroplacental blood flow, which, in turn, deteriorates the availability of oxygen and substrates to the fetus and, therefore, slows its growth trajectory [6].

A hypoxic state leads the fetal cardiovascular and metabolic adaptations to decrease metabolic rate, and the low oxygen availability leads to reduced glucose consumption compared to glucose delivery [7]. This intrauterine condition increases the risks of having cardiovascular and metabolic pathologies for the FGR fetus [8,9].

FGR are classified in early or late form if diagnosed before or after 32 weeks of gestation [10]. Late-FGR (70–80%) are characterized by a hypoxic condition caused by chronic inflammation and oxidative stress. This results in late placental alterations and modifications in cerebral–placental ratio [11]. On the contrary, early-FGR (20–30%) are mainly associated with placental hypoperfusion, leading to a condition of chronic fetal hypoxia.

SGA (small for gestational age) infants are defined with birthweight less than the 10th percentile for the gestational age [12]. For a long time, SGA has been used as a synonym for FGR. Although there is some overlap between SGA and FGR, it is now widely recognized that the two terms refer to different conditions. FGR is a pathologic condition where the fetus is deprived of oxygen and nutrients, whereas approximately 40% of fetuses with a fetal size less than the 10th percentile are constitutionally small and healthy [13].

Despite this, SGA infants may have an increased risk of perinatal morbidity and mortality, especially if undiagnosed before birth, and adverse long-term outcomes such as cardiovascular disease or poor cognitive development in adulthood [14,15,16].

Early identification of FGR and SGA conditions, together with an optimization of delivery times, reduce the fetal morbidity and mortality of the fetuses themselves.

Few studies have compared SGA and FGR, while most studies have addressed adverse perinatal and neonatal outcomes in FGR fetuses in relation to gestational age at birth.

In this study we evaluated the perinatal outcomes unrelated to gestational age of undetected SGA, born at term. Secondly, we compared perinatal and neonatal birth outcomes in relation to intrauterine pattern of fetal growth, according to severity.

## 2. Materials and Methods

This retrospective cohort study analyzed the data collected in the maternal-fetal medicine center of the Buzzi Hospital in Milan, between February 2010 and May 2021. Data were recorded through an internal multimedia database, filled out at the end of delivery by the health care providers, and ultrasound reporting program ViewPoint 5.6.21.12 (ViewPoint Bildverarbeitung GmbH, Weßling, Germany).

The total sample was composed by 906 singleton pregnancies, collected through an ultrasound dating based on crown-rump length (CRL) before 12 weeks, with no fetal malformations or chromosomal abnormalities: 655 pregnancies with appropriate neonatal weight for gestational age (AGA), 189 pregnancies complicated by fetal growth restriction (FGR), and 62 pregnancies with small neonatal weight for gestational age (SGA).

AGA infants were born from spontaneous and single pregnancies characterized by the absence of maternal and fetal pathologies. The analysis included women aged from 18 to 45 years, who delivered at term with spontaneous onset of labor or caesarean section before labor.

Fetal growth restriction has been defined according to diagnostics criteria by Delphi procedure [10] and consequently subdivided in 132 early-FGR and 57 late-FGR. EFW (estimated fetal weight) was calculated considering head circumference (HC), abdominal circumference (AC), and femur length (FL) using the Hadlock formula (HC-AC-FL). The reference percentiles for AC and EFW were those reported by Nicolini et al. [17] and Marsala et al. [18], respectively. Pregnancies complicated by fetal growth restriction were followed up in our department of maternal fetal pathology according to our clinical protocol.

The SGA group was composed of infants with a birthweight less than the 10th percentile according to INeS (Italian Neonatal Study) charts, that were not recognized as small for gestational age during pregnancy or prior to delivery.

Maternal characteristics, mode of conception, weight gain during pregnancy, and the onset of any obstetric pathologies were analyzed in all groups. Additionally, timing and mode of delivery, infant and placental weights, Apgar scores at 1 and 5 min after birth, and umbilical arterial blood gas and acid-base values (pH, base excess, lactate by RAPIDPoint^®^ 500e as a blood gas analyzer) have been collected and recorded.

Moreover, short-term infant outcomes have been assessed: ventilatory assistance in the delivery room, admission to the neonatal intensive care unit, days of hospitalization, days to achieve full enteral feeding, and onset of major neonatal complications (RDS, apnea, necrotizing enterocolitis, intraventricular hemorrhage, sepsis, jaundice, anemia, and transient hypoglycemia).

Birthweight was expressed in percentiles according to the neonatal reference INeS charts, separated by gestational age at delivery, fetal sex, and firstborn and no firstborn [12].

To compare the four groups, maternal weight gain has been adjusted for gestational age at delivery. Furthermore, the fetal–placental weight ratio was computed.

Analyses were performed using the statistical package SPSS, v.27 (IBM; Armonk, NY, USA).

Continuous variables were expressed as median [interquartile range, IQR]. Discrete variables were expressed as percentages. The data distribution of continuous variables was assessed by the Kolmogorov–Smirnov test. Continuous variables displayed a non-parametric and were thus compared among study groups by the Kruskal–Wallis test. In post hoc analyses, the Mann–Whitney U test was used. The Chi-square test was chosen for discrete variables.

Statistical significance was considered progressively greater with *p* < 0.05, *p* < 0.01, and *p* < 0.001.

## 3. Results

### 3.1. Maternal Data

Table 1 presents maternal characteristics of all groups. The pregnant women were primarily Caucasian, with average maternal age similar in all groups. In both FGR and SGA, the average pregestational maternal BMI (Body Mass Index) was statistically lower than for AGA (AGA 24.7 [IQR 22.5–27.3]) (early-FGR 22.3 [IQR 19.1–25.4] *p* < 0.001 vs. AGA; late-FGR 19.8 [IQR 18.5–22.8] *p* < 0.001 vs. AGA; SGA 20.7 [IQR 19.1–23.3] *p* < 0.001 vs. AGA) (Figure 1). The odds of being SGA, early-FGR, or late-FGR was greater in the fetuses of mothers with a lower BMI (Table 2).

Gestational age-adjusted weight gain was significantly reduced in early-FGR and SGA compared to AGA (AGA 282.1 g [IQR 236.8–368.4], early-FGR group is 258.3 g [IQR 95.3–308.4], late-FGR 263.2 g [194.4–333.3], and SGA 259.8 g [IQR 205.1–325] (early-FGR *p* < 0.001 vs. AGA; SGA *p* < 0.05 vs. AGA).

Hypertensive disorders in pregnancy occurred in 33% of early-FGR, 10% of late-FGR, and 6% of SGA in agreement with the severity of the intrauterine growth abnormality.

### 3.2. Perinatal Data

Table 3 presents perinatal data of the studied groups. Gestational week at delivery was similar in AGA (39 [IQR 39–40]) and SGA (39 [IQR 38–40]) groups and significantly lower in relation to severity in FGR (early-FGR 32 [IQR 29–38] *p* < 0.001 vs. AGA, late-FGR, and SGA; late-FGR 38 [IQR 36–39] *p* < 0.001 vs. AGA and SGA) (Table 3).

The percentage of caesarean sections was similar between AGA and SGA (18%). In late-FGR (30%), we observed a higher percentage of caesarean section than in AGA (*p* < 0.05). Additionally, cesarean section was significantly higher in early-FGR (72%) than in the other groups (*p* < 0.001).

Median birthweight was correlated with clinical severity: AGA 3330 gr [IQR 3065–3560], early-FGR 1252 gr [IQR 930–2260], late-FGR 2420 gr [IQR 2052–2620], and SGA 2690 gr [IQR 2557–2806] (*p* < 0.001 for all group vs. AGA). In fetuses with altered fetal growth, we observed significantly lower birthweight percentiles than for AGA (AGA 53 [IQR 52–54], early-FGR 5 [IQR 3–10] *p* < 0.001 vs. AGA, late-FGR 5 [IQR 3–10] *p* < 0.001 vs. AGA, SGA 5 [IQR 3–7.2] *p* < 0.001 vs. AGA).

Placental weight was significantly reduced according to the degree of severity: AGA 560 gr [IQR 500–630]; early-FGR 280 gr [IQR 200–401.2] *p* < 0.001 vs. AGA, vs. SGA; late-FGR 400 gr [IQR 330–440] *p* < 0.001 vs. AGA; SGA 440 [IQR 405–500] gr *p* < 0.001 vs. AGA. The fetal–placental weight ratio in late-FGR (5.9 [IQR 5.5–6.4]) and SGA 5.9 [5.2–6.6]) was similar to AGA (5.9 [IQR 5.4–6.4]), whereas in early-FGR it was significantly lower (early-FGR 5.2 [3.9–6.0] *p* < 0.001 vs. AGA, vs. late-FGR, vs. SGA).

The percentage of Apgar score <7 at 1 min was significantly higher in both FGR groups than in AGA (AGA 0.8%; early-FGR 33% *p* < 0.001 vs. AGA, late-FGR, SGA; late-FGR 5.3% *p* < 0.001 vs. AGA; SGA 3.2%). The percent of Apgar score <7 at 5 min was significantly higher than AGA only in early-FGR (3% *p* < 0.001 vs. AGA 0.2%). No SGA or late-FGR fetuses had an Apgar score <7 at 5 min.

Values of oxygenation and acid-base balance at delivery in FGR fetuses (Table 4) were comparable to that of AGA fetuses, while SGA fetuses showed significantly lower BE (base excess) than the other groups (AGA −3.3 [IQR −5.4–1.1], early-FGR −2.8 [IQR −5.6–1], late-FGR −3.4 [IQR −5–−2.5], SGA −5.0 [IQR −7.1–−2.8]) (SGA *p* < 0.001 vs. AGA and early-FGR, *p* < 0.05 vs. late-FGR). Similarly, in SGA, umbilical arterial lactate concentration was significantly higher than in the other groups (AGA 3.7 [IQR 2.4–5.3], early-FGR 3.0 [IQR 2.1–4.6], late-FGR 3.5 [IQR 2.6–4.7], SGA 4.4 [IQR 2.7–5.5] *p* < 0.05 vs. AGA, *p* < 0.01 vs. early-FGR). The percentage of SGA (4.8%) fetuses with pH < 7.10 was higher but not statistically significant compared with the other groups (AGA 2.7%, early-FGR 0.8 %, late-FGR 3.5%).

The differences found in fetal oxygenation remained significant regardless of the mode of delivery (Appendix A).

Table 5 presents data on neonatal complications. Access to neonatal intensive care unit (NICU) was correlated to gestational age at delivery (early-FGR 64% *p* < 0.01 vs. late-FGR 44%, *p* < 0.001 vs. SGA 11%, no AGA).

In early-FGR, the days of hospitalization in NICU were significantly higher than late-FGR and SGA, with greater difficulties in adapting to extra-uterine life. Indeed, it is the group with a higher complications rate, reported in Table 5. We have only one case of neonatal death that occurred in the early-FGR group.

## 4. Discussion

Alterations in intrauterine growth increase the risks of adverse perinatal outcomes; therefore, their correct identification is important [19,20]. In agreement with previous reports, our results show that the important maternal risk factors for reduced fetal growth are represented by age (higher in early-FGR), being primiparous (higher frequency in FGR and SGA), pregestational weight and BMI (lower in both FGR and SGA), and gestational weight gain (reduced in early-FGR) [21,22].

In our population, SGA had worse oxygenation and acid-base balance at delivery, but neonatal outcomes were not different from AGA. The risks of adverse neonatal outcomes were instead significantly increased in the FGR group [23] with the higher need of NICU access and hospitalization days strictly related to prematurity. However, FGR fetuses showed pH values in the normal range, suggesting an optimal delivery timing and mode. Prematurity-dependent neonatal outcomes were consistent with previous studies.

Since the SGA fetuses were undetected during pregnancy, they were managed as AGA fetuses with the same timing and mode of delivery. Interestingly, SGA showed higher lactates and lower base excess values compared to the other groups, probably due to a different adaptation to the stress of labor. Despite this, the gestational age and birthweight in the SGA group likely played an important role in reducing the occurrence of neonatal complications.

In our study, we focused on the worsening stress adaptation of labor for SGA infants, resulting in hyperlactacidemic and hypoxemic states at birth. A recent study has shown that SGA and FGR fetuses are adversely associated with cognitive development compared with AGA, with no significant cognitive difference between them [15]. Being small-size fetuses does not necessarily reflect pathology but puts the fetuses at higher risk of adverse outcomes. While there is agreement about the clinical management of early and late-FGR [24], detection of SGA in the weeks prior to term is still a challenge. Indeed, a gold standard to detect small for gestational age fetuses is lacking. Currently, in Italy, symphysis-fundal height (SFH) measurements and third-trimester ultrasound (28–32 weeks) are used to monitor fetal growth, although routine ultrasound is not current practice because it is not included in national essential levels of care. Recent studies have shown that ultrasound at 36 weeks is more effective in detecting SGA fetuses than ultrasound at 32 weeks [25]; however, this choice would increase the risk of not detecting early-FGR fetuses. There is not sufficient evidence to determine whether SFH measurement is effective in detecting alterations of fetal growth. However, SFH is a simple and inexpensive clinical activity and may constitute a first level screen [26,27]. Many studies have supported the use of the customized charts to screen for fetal growth alterations. The customized approach detected more SGA neonates when compared with the IG-21 (INTERGROWTH-21st) project standard, but it was less specific than the latter. Both standards were poor at predicting SGA neonates at risk for short-term adverse outcomes [28].

In our population, low maternal pregestational BMI was a significant risk factor for having a small baby, both SGA as well as early and late-FGR. Recently, most studies have stressed the relevance of maternal obesity as a risk factor during pregnancy. However, low BMI is also associated with adverse obstetric outcomes [3], and appropriate nutritional counseling should, therefore, be considered for these pregnancies. Moreover, low maternal BMI together with low maternal weight gain during pregnancy is associated with potential eating disorders that increase the risk of low micronutrients intake and status [29]. Specifically, low maternal iron stores have been reported in mothers with low BMI, with an increased risk of SGA and decreased neurocognitive function in the offspring [30].

### Strengths and Limitations

The data were collected retrospectively, introducing the potential for bias. Besides this, the study is limited to one center and involved a homogenous population in a large metropolitan area. Nonetheless, this study represents one of the few studies analyzing smallness as a risk factor of perinatal and neonatal adverse outcome in a large pregnant population.

In conclusion, undetected SGA fetuses present worse oxygenation and acid base balance at delivery than AGA fetuses. However, differently from FGR fetuses, undetected SGA fetuses did not show worse immediate neonatal outcomes than AGA. The timing of the routine third-trimester ultrasound scan together with the utilization of SFH evaluation should be rethought in order to have more active surveillance around delivery for the detection of alterations of fetal growth, particularly in women with low BMI and low gestational weight gain.

## Figures and Tables

**Figure 1 jcm-11-02729-f001:**
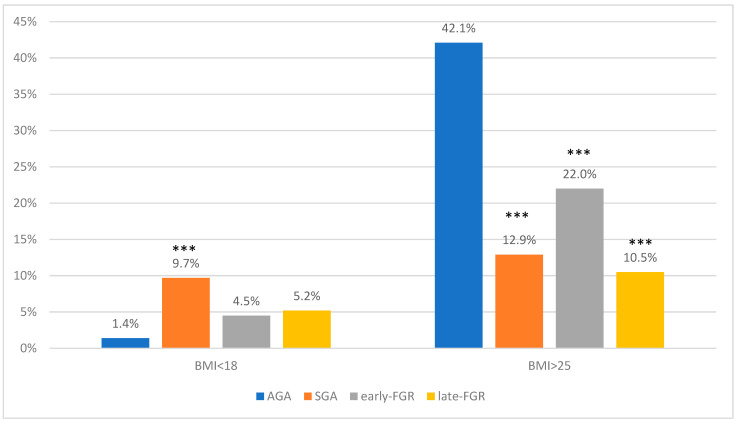
Maternal BMI < 18 or >25 in AGA, SGA, early-FGR, and late-FGR groups. *** *p* value < 0.001 vs. AGA.

**Table 1 jcm-11-02729-t001:** Maternal characteristics.

Features	AGA (N = 655)	SGA (N = 62)	Early-FGR (N = 132)	Late-FGR (N = 57)
*Age* (yrs)	33.0 (30–36)	33.0 (30–37)	35.0 (30–38)	33.0 (30–36)

*Pregestational weight* (kg)	67.0 (60–74)	56.0 (50–60)	60.0 (52–70)	54.0 (41–67)
		***, ^++^	***	***, ^+^
*BMI* (kg/m²)	24.7 (22.5–27.3)	20.7 (19.1–23.3)	22.3 (19.1–25.4)	19.8 (18.5–22.8)
		***, ^+^	***	***, ^++^
<18 kg/m²	9 (1.4%)	6 (9.7%)	6 (4.5%)	3 (5.2%)
		***		
>25 kg/m²	267 (42%)	8 (12.9%)	29 (22%)	6 (10.5%)
		***	***	***
*Weight gain* (kg)	11.0 (6–16)	10.0 (8–13)	8.5 (7–11.5)	10.0 (8–13)
			***, ^^^	*
*Weight gain/gestational age at delivery* (g/week)	282.1 (236.8–368.4)	259.8 (205.1–325) *	258.3 (95.3–308.4) ***	263.2(194.4–333.3)
*Pregnancy*				
*Primiparous*	306 (47%)	39 (63%)	98 (78%)	37 (67%)
		*	***, ^^^	*
*Multiparous*	349 (53%)	23 (37%)	28 (22%)	18 (33%)
*Conception*				
*ART*	0	1 (2%)	11 (9%)	4 (8%)
*Spontaneous*	655 (100%)	56 (98%)	119 (91%)	52 (92%)
*Ethnicity*				
*Caucasian*	500 (77%)	102 (79%)	102 (80%)	49 (87%)
*African*	11 (2%)	0	6 (5%)	-
*Asian*	65 (10%)	4 (6.5%)	10 (8%)	1 (2%)
*Middle Eastern*	30 (5%)	4 (6.5%)	4 (3%)	4 (7%)
*South American*	38 (6%)	5 (8%)	5 (4%)	2 (4%)

Continuous variables are expressed as median (interquartile range). All the other features are expressed as categorical variables in frequencies and their percentage in the brackets. AGA: Appropriate for Gestational Age. SGA: Small for Gestational Age. FGR: Fetal Growth Restriction. ART: Assisted Reproductive Techniques. BMI: Body Mass Index. * *p* value < 0.05 vs. AGA; *** *p* value < 0.001 vs. AGA. ^^^
*p* value < 0.05 vs. SGA. ^+^
*p* value < 0.05 vs. early-FGR; ^++^
*p* value < 0.01 vs. early-FGR.

**Table 2 jcm-11-02729-t002:** Odds ratios (OR) of maternal BMI in SGA, early-FGR, and late-FGR fetuses.

Groups	BMI < 18	BMI > 25
	*OR (95% CI:); p **	*OR (95% CI:); p **
*SGA*	OR 3.5 (1.57–7.77); 0.002	OR 0.14 (0.69–0.28); 0.0001
*early-FGR*	OR 3.48 (1.7–7.04); 0.0005	OR 0.38 (0.25–0.59); 0.0001
*late-FGR*	OR 4.88 (2.25–10.6); 0.0001	OR 0.11 (0.05–0.26); 0.001

CI: confidence intervals. * *p*-value < 0.05 is considered significant.

**Table 3 jcm-11-02729-t003:** Data at delivery.

Features	AGA (N = 655)	SGA (N = 62)	Early-FGR (N = 132)	Late-FGR (N = 57)
*Gestational age* (weeks)	39 (39–40)	39 (38–40)	32 (29–38)	38 (36–39)
			***, ^###^, ^^^^^	***, ^^^^^
*Time from diagnosis to delivery* (weeks)	0	0	9 (3–15) ^###^	3.5 (2–5)
*Sex*				
*Male*	318 (52%)	38 (61%)	46 (35%)	25 (44%)
			***, ^^^^^	
*Female*	292 (48%)	24 (39%)	86 (65%) ***, ^^^^^	32 (56%)
*Delivery mode*				
*Cesarean section*	117 (18%)	11 (18%)	95 (72%)	17 (30%)
			***, ^###^, ^^^^^	*
*Vaginal delivery*	538 (82%)	51 (82%)	37 (28%)	40 (70%)
*Birthweight (gr)*	3330 (3065–3560)	2690 (2557–2806)	1252 (930–2260)	2420 (2052–2620)
		***	***, ^^^^^	***
*Birthweight (percentile)*	53 (52–54)	5 (3–7.2)	5 (3–10)	5 (3–10)
		***	***	***
*Placental weight (gr)*	560 (500–630)	440 (405–500)	280 (200–401.2)	400 (330–440)
		***	***, ^^^^^	***
*Fetal/Placental weight*	5.9 (5.4–6.4)	5.9 (5.2–6.6)	5.2 (3.9–6.0) ***, ^###^, ^^^^^	5.9 (5.5–6.4)

Continuous variables are expressed as median (interquartile range). All the other features are expressed as categorical variables in frequencies and their percentage in the brackets. * *p* value < 0.05 vs. AGA; *** *p* value < 0.001 vs. AGA. ^^^^^
*p* value < 0.001 vs. SGA. ^###^
*p* value < 0.001 vs. late-FGR.

**Table 4 jcm-11-02729-t004:** Delivery outcomes.

Features	AGA (N = 655)	SGA (N = 62)	Early-FGR (N = 132)	Late-FGR (N = 57)
*Apgar < 7 1′*	5 (0.8%)	2 (3.2%)	44 (33%)	3 (5.3%)
			***, ^###^, ^^^^^	***
*Apgar < 7 5′*	1 (0.2%)	0	4 (3%)	0
			***	
*pH < 7.10*	18 (2.7%)	3 (4.8%)	1 (0.8%)	2 (3.5%)
*BE*	−3.3 (−5.4–1.1)	−5.0 (−7.1–−2.8)	−2.8 (−5.6–1.0)	−3.4 (−5–−2.51)
		***, ^+++^, ^#^		
*Lactate*	3.7 (2.4–5.3)	4.4 (2.7–5.5)	3.0 (2.1–4.6)	3.5 (2.6–4.7)
		*, ^++^		

Continuous variables are expressed as median (interquartile range). All the other features are expressed as categorical variables in frequencies and their percentage in the brackets. BE: base excess. * *p* value < 0.05 vs. AGA; *** *p* value < 0.001 vs. AGA. ^^^^^
*p* value < 0.001 vs. SGA. ^#^ *p* value < 0.05 vs. late-FGR. ^###^
*p* value < 0.001 vs. late-FGR. ^++^
*p* value < 0.01 vs. early-FGR; ^+++^
*p* value < 0.001 vs. early-FGR.

**Table 5 jcm-11-02729-t005:** Neonatal complications.

Features	SGA (N = 62)	Early-FGR (N = 132)	Late-FGR (N = 57)
*Access in NICU*	7 (11%)	84 (64%)	25 (44%)
		^##^, ^^^^^	^^^^^
*Days in NICU (days)*	3 (2–4)	28 (4–52)	4 (3–5)
		^##^, ^^^^	^^^^
*Jaundice*	4 (6.4%)	51 (38.6%)	4 (7%)
		^###^, ^^^^^	
*Intraventricular Hemorrhage*	0	1 (0.8%)	1 (1.7%)
*Anemia*	0	37 (28%)	1 (1.7%)
		^###^	
*Hypoglycemia*	6 (9.7%)	17 (13%)	9 (16%)
*Full enteral feeding (days)*	0	10 (1–19) ^###^	1 (0.25–1.75)
*RDS*	1 (1.6%)	56 (42%)	4 (7%)
		^###^, ^^^^^	
*Ventilatory-assistance*	2 (3.2%)	61 (46%) ^###^, ^^^^^	7 (12%)
*Apneas*	1 (1.6%)	22 (17%) ^#^, ^^^^^	2 (3.5%)
*NEC*	0	6 (4.5%)	0
*Sepsis*	0	11 (8.3%)	1 (1.7%)
*Fetal death*	0	1 (0.8%)	0

Continuous variables are expressed as median (interquartile range). All the other features are expressed as categorical variables in frequencies and their percentage in the brackets. NICU: Neonatal Intensive Care Unit. RDS: Respiratory Distress Syndrome. NEC: Necrotizing Enterocolitis. ^^^^
*p* value < 0.01 vs. SGA; ^^^^^
*p* value < 0.001 vs. SGA. ^#^
*p* value < 0.05 vs. late-FGR; ^##^
*p* value < 0.01 vs. late-FGR; ^###^
*p* value < 0.001 vs. late-FGR.

## Data Availability

Data sharing not applicable.

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
