# Peer review of "Perinatal and Neonatal Outcomes in Fetal Growth Restriction and Small for Gestational Age"

_jcm, 2022, doi:10.3390/jcm11102729_

Round 1

Reviewer 1 Report

The research is well-designed, and I congratulate the authors for the extensive and objective investigation of pregnant women, fetuses, and newborns/infants.

Main concerns:

The abstract is missing.

Introduction section: could be improved, by providing info regarding the importance of the studied feto-maternal and neonatal features in your research. Also, previous findings and the originality of the study should be somehow presented.

Abbreviations should be explained (INeS, Methods section).

The Discussion section should enclose a proposal for optimal management in the third trimester, especially regarding SGA fetuses, based on the study results, and combined with literature data.

The interpretation of the research data compared to available literature is missing in the Discussion section.

“Strengths and limitations” considerations should be added

The Conclusion section is missing.

Author Response

Response to Reviewers' comments

First, we would like to thank all the Reviewers for their constructive and useful comments. We have addressed all the raised points and we feel that the manuscript has been significantly improved.

Here below are point-to-point answers:

First reviewer

The abstract is missing.

The abstract was uploaded in the “Abstract Box”. Now we have included it in the article.

Introduction section: could be improved, by providing info regarding the importance of the studied feto-maternal and neonatal features in your research. Also, previous findings and the originality of the study should be somehow presented.

Thank you. We have modified the text following your advice on lines 22-28.

Abbreviations should be explained (INeS, Methods section).

 We explained this on line 90.

The Discussion section should enclose a proposal for optimal management in the third trimester, especially regarding SGA fetuses, based on the study results, and combined with literature data.

We thank the Reviewer for this suggestion. We have now included a proposal for optimal management on lines 221-224 of the Discussion.

The interpretation of the research data compared to available literature is missing in the Discussion section.

The Discussion has been expanded to compare obtained results more deeply with the available literature on lines 170-211.

“Strengths and limitations” considerations should be added

We added strengths and limitations on lines 213-217.

The Conclusion section is missing

We included the conclusions at the end of the discussion on lines 219-224.

Reviewer 2 Report

In the study “Perinatal and neonatal outcomes in fetal growth restriction and small for gestational age” authors presented a retrospective local study considering deliveries during a one-year period. They classified deliveries in AGA, SGA (diagnosed at delivery), early-FGR and late-FGR. Thy demonstrated that FGR have adverse perinatal outcomes, mainly early-FGR. The study needs a full revision by a native English speaker to improve several grammatical mistakes. There are some minor and major concerns that I would like to share:

Minor comments:

Introduction, third paragraph, I suggest to modify “FGR” by “FGR secondary to placental insufficiency”.

Material and Methods, second paragraph, modify “906 single pregnancies” with “906 singleton pregnancies”.

Sub section “Neonatal outcomes”, first paragraph, please modify “(tab 2)” for “(Table 2)”.

Major comments:

Statistical analysis. Authors mentioned that continuous variables were compared using t-student test. However, first the variables distribution should be assessed with another test (i.e. Shapiro Wilk, Kolmogorov-Smirnov or Shapiro-Francia). If a normal distribution is obtained, then a parametric test should be used. T-test is for comparison of normally distribution variables comparing two groups. If more than 2 groups will be compared (as in this study), then an ANOVA test with post-hoc comparison test (i.e. Bonferoni) within pairs should be used and expressed as mean (standard deviation). If a non-parametric distribution is observed, then a Kruskal-Wallis test should be used and expressed as median (interquartile range). Please check the distributions of all your continuous variables and modify the analyses and results accordingly. I recommend to authors that in a way to demonstrate that SGA and FGR population have a higher risk of several perinatal outcomes, authors should estimate Odds Ratios with a logistic regression analysis.

Results. In Table 1, Conception, what is MAP? All abbreviations should be added at the bottom of the table. In results section, if there is a Table, there is no need to write all the results within the table in the main document. It is better to bring the most important idea to the main section, otherwise, the table is not needed. Please modify this section according to this suggestion.

In Table 3, APGAR score is not a continuous variable (there is no APGAR 3.2 score), is a discrete variable, therefore, the analysis should consider that and expressed as a non-decimal number or considered as a dichotomic variable (less than 7 [yes/no]). In this line, what was the rate of a low APGAR in all the groups? The same question for pH values less than 7.1 in cord blood.

Graph 1 presents the same results as in Table 1, therefore, it is not necessary to show duplicate results. I recommend to estimate the OR for early-FGR, late-FGR and SGA, considering AGA as a reference, when BMI is less than 18 and more than 30.

In graph 2, I suggest that placental weight and neonatal weight have different symbols (i.e. an X for placental weight and a circle for neonatal weight). What is the porpoise of this figure? Maybe it could be better to create a graph of the feto/placental ratio in all groups.

Despite of an adequate sample size in sub groups, all results are according to what is expected in these populations. Obviously early-FGR have the worst perinatal outcome, then late-FGR. I think that the analysis should considered, within FGR population, which fetal assessment has a better correlation with adverse outcomes: umbilical artery percentile, ductus venosus status, brain sparring, estimated fetal percentile, gestational age at diagnosis, presence of preeclampsia, etc. It is important to bring an interesting or different analysis, otherwise the present draft is presenting no new evidence that could be of interest for readers.

Discussion section needs a better development. It is not structured. I recommend authors to write a first paragraph that brings the main message of the study, then 2 or 3 paragraphs contrasting your results with other studies already published. Then another paragraph of strengths and weaknesses of the study, and finally a conclusion paragraph.

Author Response

Response to Reviewers' comments

First, we would like to thank all the Reviewers for their constructive and useful comments. We have addressed all the raised points and we feel that the manuscript has been significantly improved.

Here below are point-to-point answers:

Introduction, third paragraph, I suggest to modify “FGR” by “FGR secondary to placental insufficiency”.

We modified FGR as you suggest on line 38.

Material and Methods, second paragraph, modify “906 single pregnancies” with “906 singleton pregnancies”.

We modified this on line 74.

Sub section “Neonatal outcomes”, first paragraph, please modify “(tab 2)” for “(Table 2)”.

We modified Table 2 on lines 129 and 132.

Statistical analysis. Authors mentioned that continuous variables were compared using t-student test. However, first the variables distribution should be assessed with another test (i.e. Shapiro Wilk, Kolmogorov-Smirnov or Shapiro-Francia). If a normal distribution is obtained, then a parametric test should be used. T-test is for comparison of normally distribution variables comparing two groups. If more than 2 groups will be compared (as in this study), then an ANOVA test with post-hoc comparison test (i.e. Bonferoni) within pairs should be used and expressed as mean (standard deviation). If a non-parametric distribution is observed, then a Kruskal-Wallis test should be used and expressed as median (interquartile range). Please check the distributions of all your continuous variables and modify the analyses and results accordingly. I recommend to authors that in a way to demonstrate that SGA and FGR population have a higher risk of several perinatal outcomes, authors should estimate Odds Ratios with a logistic regression analysis.

We thank the Reviewer for this comment. We have now modified the Statistical analysis following your advice (lines 105-111). The data distribution of continuous variables was assessed by the Kolmogorov–Smirnov test. Continuous variables displayed a non-parametric distribution and were thus compared among study groups by the Kruskal-Wallis test. In post hoc analyses, the Mann–Whitney U test was used. Continuous variables were expressed as median [interquartile range, IQR].

We estimated the Odds Ratio of BMI. (Lines 120-121)

Results. In Table 1, Conception, what is MAP? All abbreviations should be added at the bottom of the table. In results section, if there is a Table, there is no need to write all the results within the table in the main document. It is better to bring the most important idea to the main section, otherwise, the table is not needed. Please modify this section according to this suggestion.

We specified MAP definition (medically assisted procreation) on line 257.

In Table 3, APGAR score is not a continuous variable (there is no APGAR 3.2 score), is a discrete variable, therefore, the analysis should consider that and expressed as a non-decimal number or considered as a dichotomic variable (less than 7 [yes/no]). In this line, what was the rate of a low APGAR in all the groups? The same question for pH values less than 7.1 in cord blood.

Thank you, we have modified the text following your advice. Apgar score < 7 is now presented on lines 146-150, while pH < 7.10 is presented on lines 157-158.

Graph 1 presents the same results as in Table 1, therefore, it is not necessary to show duplicate results. I recommend to estimate the OR for early-FGR, late-FGR and SGA, considering AGA as a reference, when BMI is less than 18 and more than 30.

We chose to create a graph of BMI because it is an important risk factor for altered fetal growth and we wanted to make it more readily apparent. We estimated the OR using BMI < 18 and BMI > 25, as we considered in the graph 1.

In graph 2, I suggest that placental weight and neonatal weight have different symbols (i.e. an X for placental weight and a circle for neonatal weight). What is the porpoise of this figure? Maybe it could be better to create a graph of the feto/placental ratio in all groups.

In according with your suggestion, we preferred to delete Graph 2 because it did not add any more information to Table 2.

Despite of an adequate sample size in sub groups, all results are according to what is expected in these populations. Obviously early-FGR have the worst perinatal outcome, then late-FGR. I think that the analysis should considered, within FGR population, which fetal assessment has a better correlation with adverse outcomes: umbilical artery percentile, ductus venosus status, brain sparring, estimated fetal percentile, gestational age at diagnosis, presence of preeclampsia, etc. It is important to bring an interesting or different analysis, otherwise the present draft is presenting no new evidence that could be of interest for readers.

All FGR fetuses are followed according to our clinical protocol and their delivery time is detected in relation to both ultrasound and computerized CTG criteria. In this study, we’ve chosen not to show ultrasonographic data in order to focus on unexpected SGA outcomes.

Discussion section needs a better development. It is not structured. I recommend authors to write a first paragraph that brings the main message of the study, then 2 or 3 paragraphs contrasting your results with other studies already published. Then another paragraph of strengths and weaknesses of the study, and finally a conclusion paragraph

We have modified the discussion section on lines 170-224.

Reviewer 3 Report

The authors present data and their conclusions on postpartum outcomes in AGA newborns compared to 3 subgroups of small babies i.e. SGA, early-FGR and late-FGR. This study may add valuable information to current knowledge if several improvements are applied as suggested below.

It would be very helpful for reviewing if the manuscript is prepared more strictly to the rules e.g. with numbers of lines.

Citations from the beginning should be corrected as numeration is confusing.

Consider more citations of other relevant authors.

Clarify why SGA group was constructed using after birth data instead of based on prenatal ultrasound parameters derived from your View Point database.

MAP needs to be described more detailed whether it is IVF, ovulation induction, AIH/AID etc.

Did you check smoking incidence in your groups? Any other substances affecting fetal growth like alcohol, drugs?

What is the incidence of hypertension in AGA? Provide references for classification of hypertension in pregnancy applied in your study.

Graph 2 should be reconstructed to show the results more precisely e.g. as color lines of trends separately for birthweight and placental mass in each group instead of points.

Section "Neonatal outcome" in Apgar 1 description needs to be corrected as SGA is compared to SGA.

How can you explain differences in Apgar scores between groups compared to lack of statistical changes in cord blood pH?

How can you explain higher cord blood pH in early-FGR?

EB abbr should be changed to BE in all tables.

Bicarbonates and fetal Hb evaluations are listed in the study protocol but missing in the presentation. Clarify

Discussion is too short and not referred to earlier studies especially in its second part. Needs to be extended.

Explain how you propose to identify SGA fetuses prenatally to reduce postpartum lower pH and higher BE.

Reference section must be corrected, some citations are incomplete.

Author Response

Response to Reviewers' comments

First, we would like to thank all the Reviewers for their constructive and useful comments. We have addressed all the raised points and we feel that the manuscript has been significantly improved.

Here below are point-to-point answers:

It would be very helpful for reviewing if the manuscript is prepared more strictly to the rules e.g. with numbers of lines.

We’ve prepared the manuscript according to your suggestion.

Citations from the beginning should be corrected as numeration is confusing.

We’ve corrected numeration of citations.

Consider more citations of other relevant authors.

We added more citations.

Citations:

  1. Damhuis, S. E., Ganzevoort, W. & Gordijn, S. J. Abnormal Fetal Growth Small for Gestational Age, Fetal Growth Restriction, Large for Gestational Age: Definitions and Epidemiology. doi:10.1016/j.ogc.2021.02.002. (On line 55)

  1. Pallotto, E. K. & Kilbride, H. W. Perinatal Outcome and Later Implications of Intrauterine Growth Restriction. (On line 58)

  1. Sacchi, C. et al. Association of Intrauterine Growth Restriction and Small for Gestational Age Status With Childhood Cognitive Outcomes: A Systematic Review and Meta-analysis. JAMA Pediatrics 174, 1 (2020). (On line 58)

  1. Henrichs, J. et al. Cost-Effectiveness of Routine Third Trimester Ultrasound Screening for Fetal Growth Restriction Compared to Care as Usual in Low-Risk Pregnancies: A Pragmatic Nationwide Stepped-Wedge Cluster-Randomized Trial in The Netherlands (the IRIS Study). International Journal of Environmental Research and Public Health 2022, Vol. 19, Page 3312 19, 3312 (2022). (On line 58)

  1. Doctor, B. A., O’Riordan, M. A., Kirchner, H. L., Shah, D. & Hack, M. Perinatal correlates and neonatal outcomes of small for gestational age infants born at term gestation. Am J Obstet Gynecol 185, 652–659 (2001). (On line 174)

  1. Larkin, J. C., Chauhan, S. P. & Simhan, H. N. Small for Gestational Age: The Differential Mortality, When Detected versus Undetected Antenatally. in American Journal of Perinatology vol. 34 409–414 (Thieme Medical Publishers, Inc., 2017). (On line 177)

  1. Lees, C. C. et al. Clinical Opinion: The diagnosis and management of suspected fetal growth restriction: an evidence-based approach. Am J Obstet Gynecol 226, 366–378 (2022). (On line 191)

  1. Martín-Palumbo, G., Atanasova, V. B., Rego Tejeda, M. T., Antolín Alvarado, E. & Bartha, J. L. Third trimester ultrasound estimated fetal weight for increasing prenatal prediction of small-for-gestational age newborns in low-risk pregnant women. J Matern Fetal Neonatal Med (2021) doi:10.1080/14767058.2021.1920915. (On line 196)

  1. Robert Peter, J., Ho, J. J., Valliapan, J. & Sivasangari, S. Symphysial fundal height (SFH) measurement in pregnancy for detecting abnormal fetal growth. Cochrane Database of Systematic Reviews 2015, (2015). (On line 199)

  1. Papageorghiou, A. et al. International standards for symphysis-fundal height based on serial measurements from the Fetal Growth Longitudinal Study of the INTERGROWTH-21st Project: prospective cohort study in eight countries On behalf of the International Fetal and Newborn Growth Consortium for the 21st Century (INTERGROWTH-21st). doi:10.1136/bmj.i5662. (On line 199)

  1. Odibo, A. O. et al. Customized fetal growth standard compared with the INTERGROWTH-21st century standard at predicting small-for-gestational-age neonates. Acta Obstet Gynecol Scand 97, 1381–1387 (2018). (On line 203)

  1. Mantel, Ä., Hirschberg, A. L. & Stephansson, O. Association of Maternal Eating Disorders With Pregnancy and Neonatal Outcomes. JAMA Psychiatry 77, 285–293 (2020). (On line 210)

  1. Koubaa, S., Hällström, T., Brismar, K., Hellström, P. M. & Hirschberg, A. L. Biomarkers of nutrition and stress in pregnant women with a history of eating disorders in relation to head circumference and neurocognitive function of the offspring. BMC Pregnancy and Childbirth 15, (2015). (On line 211)

Clarify why SGA group was constructed using after birth data instead of based on prenatal ultrasound parameters derived from your View Point database.

In our hospital we have an 8% annual rate of unexpected SGA, which has shown regular growth on routine ultrasound. The goal is to better assess undetected SGA to improve their neonatal outcomes.

MAP needs to be described more detailed whether it is IVF, ovulation induction, AIH/AID etc.

This would be very interesting, but we do not have more details about it.

Did you check smoking incidence in your groups? Any other substances affecting fetal growth like alcohol, drugs?

We do not have comprehensive data on this. An increased risk of FGR with smoking is reported in the literature. It will be interesting to analyze these data in the future.

What is the incidence of hypertension in AGA? Provide references for classification of hypertension in pregnancy applied in your study.

We have considered AGA infants born from spontaneous and single pregnancies characterized by the absence of maternal and fetal pathologies (lines 79-80)

Graph 2 should be reconstructed to show the results more precisely e.g. as color lines of trends separately for birthweight and placental mass in each group instead of points.

We have now preferred to delete graph 2 because it did not add any more information to Table 2.

Section "Neonatal outcome" in Apgar 1 description needs to be corrected as SGA is compared to SGA.

We corrected this on lines 146-150.

How can you explain differences in Apgar scores between groups compared to lack of statistical changes in cord blood pH?

Gestational age and birthweight in the SGA group play an important role in a lower frequency of Apgar <7. The stress of labor and lower reserves in the SGA group explain the worst blood gas values (pH, EB and lactate).

How can you explain higher cord blood pH in early-FGR?

Ph is higher in early FGR because they are mostly born by cesarean section, with reduced fetal stress related to labor.

EB abbr should be changed to BE in all tables.

We have changed this on line 289.

Bicarbonates and fetal Hb evaluations are listed in the study protocol but missing in the presentation. Clarify??

All variables mentioned (pH, BE and lactate) are shown in table 3.

Discussion is too short and not referred to earlier studies especially in its second part. Needs to be extended.

We extended the discussion on lines 170-211 and on lines 219-224. We added the strengths and limitations on lines 213-217.

Explain how you propose to identify SGA fetuses prenatally to reduce postpartum lower pH and higher BE.

We’ve added our clinical propose on lines 221-224.

Reference section must be corrected, some citations are incomplete.

We corrected the citations.

Round 2

Reviewer 2 Report

I congratulate authors for the draft improvement. I have no comments

Author Response

Thank you!

Reviewer 3 Report

Most of the corrections have been applied by the authors.

Some changes are still pending or necessary.

1/ Numeration of tables is confusing, change 1a to 2 and following numbers, put description at the top not bottom, all abbrev should be explained, "apneas" to be corrected, "Data at delivery" should be changed for more relevant description. There are 5 tables wrongly numerated.

2/ MAP abbrev is confusing with mean arterial pressure. Give the reference for your abbrev or change into ART (assisted reproductive techniques) and explain in text it is of unknown kind. Alternatively the abbrev could be "Medically Assisted Conception (MAC)".

3/ Fig 2 seems to be unnecessary, data are presented in one of 5 tables

4/ Reference 21 must be corrected.

Author Response

Thank you for your comments, please see the attachment.
